# SealNet 2.0: Human-Level Fully-Automated Pack-Ice Seal Detection in Very-High-Resolution Satellite Imagery with CNN Model Ensembles

**Bento C. Gonçalves** [1,2,*], **Michael Wethington** [1] **and Heather J. Lynch** [1,2]

1   Department of Ecology and Evolution, Stony Brook University, Stony Brook, NY 11794-5245, USA
2   Institute for Advanced Computational Sciences, Stony Brook University, Stony Brook, NY 11794-3365, USA
*   Correspondence: bento.goncalves@stonybrook.edu

**Abstract:** Pack-ice seals are key indicator species in the Southern Ocean. Their large size (2–4 m) and continent-wide distribution make them ideal candidates for monitoring programs via very-high-resolution satellite imagery. The sheer volume of imagery required, however, hampers our ability to rely on manual annotation alone. Here, we present SealNet 2.0, a fully automated approach to seal detection that couples a sea ice segmentation model to find potential seal habitats with an ensemble of semantic segmentation convolutional neural network models for seal detection. Our best ensemble attains 0.806 precision and 0.640 recall on an out-of-sample test dataset, surpassing two trained human observers. Built upon the original SealNet, it outperforms its predecessor by using annotation datasets focused on sea ice only, a comprehensive hyperparameter study leveraging substantial high-performance computing resources, and post-processing through regression head outputs and segmentation head logits at predicted seal locations. Even with a simplified version of our ensemble model, using AI predictions as a guide dramatically boosted the precision and recall of two human experts, showing potential as a training device for novice seal annotators. Like human observers, the performance of our automated approach deteriorates with terrain ruggedness, highlighting the need for statistical treatment to draw global population estimates from AI output.

**Keywords:** pack-ice seal; remote sensing; Worldview-3; Antarctica; computer vision; deep learning; instance segmentation; U-Net

## 1. Introduction

Pack-ice seals (crabeater seals (*Lobodon carcinophaga*), Weddell seals (*Leptonychotes weddelli*), leopard seals (*Hydrurga leptonyx*), and Ross seals (*Omnatophoca rossi*)) play a prominent role as key Antarctic marine predators in the Southern Ocean (SO) ecosystem [1,2]. The importance of crabeater seals as a major consumer of Antarctic krill is well documented and the focus of considerable attention vis a vis seal conservation and management [3]. Antarctic krill, in turn, is not only a key component in biogeochemical cycles in the SO through its consumption of phytoplankton [4], but also sits at the base of the SO foodweb [5], attaining upwards of 300 million tonnes of biomass during its peak abundance in the austral summer [6]. These massive concentrations of krill sustain large populations of a wide range of marine predators, from marine mammals and seabirds to fish and squid. Understanding fluctuations in krill populations through the SO, and how they track sea ice conditions [7], is fundamental to gauge the health of the SO ecosystem—especially under anthropogenic climate change [7]. Through their trophic interaction with krill, and due to the inherent difficulties of estimating krill populations directly [6,8], surveying pack-ice seal populations provides a window to krill population dynamics. Among krill predators in the SO, pack-ice seals haul out on sea ice for breeding, molting, and predator avoidance [9] and are large enough to be individually counted using high-resolution satellite imagery [10].

For these reasons, pack-ice seals may serve as useful proxies for krill abundance if they can be surveyed at large spatial scales cost-effectively.

The role of pack-ice seals as an ecosystem health barometer in the SO has fueled several population estimation efforts. The earliest efforts to do so are reviewed in [11] but a more recent and highly structured attempt at estimating pan-Antarctic pack-ice seal population abundance was organized by the Scientific Committee for Antarctic Research under the umbrella of the International Antarctic Pack Ice Seals (APIS) program [12]. APIS was a large collaboration between six countries from 1994 to 2000 that employed vessel- and helicopter-based line transects to sample seal densities at several locations and times. Though such studies brought invaluable insights on seal biology (e.g., [13–15]), the spatial and temporal coverages are limited to draw continent-wide population estimates. Aerial transect-based monitoring programs such as APIS, especially in remote locations such as the SO, are not only resource-intensive [12] but dangerous for field biologists [16], making them unfeasible as means to repeatedly survey pack-ice seal populations.

Remote sensing imagery from very-high-resolution (VHR) sensors provides a cheaper and safer alternative to survey several polar animal populations [17], and has been demonstrated to work for pack-ice seals [10] and several other large marine mammal populations (e.g., southern elephant seals [18], walrus [19], and whales [20]). Manually annotating pack-ice seals in VHR imagery, however, is not only laborious but also extremely difficult given variations in lighting, substrate conditions, and imagery post-processing artifacts. These difficulties, allied with a scarcity of trained observers and the non-negligible cost of commercial VHR imagery, hamper our efforts to survey imagery manually at scale. Citizen science efforts mitigate these limitations by crowd-sourcing large areas across several volunteers (e.g., [21]). In order to draw useful ecological insights, however, robust sampling designs with overlapping predictions are needed to offset annotator inexperience [22,23]. In an effort by LaRue and collaborators [24], the Satellite Over Seals project (SOS), a 3-year citizen science campaign with >300,000 volunteers, enabled a Weddell seal survey covering the entirety of Antarctic fast-ice VHR imagery for the year of 2011, and the most comprehensive global estimate for the species to date [25]. Without a massive scale-up, even such large-scale campaigns will fail to keep pace with the volume of imagery collected over the Southern Ocean [26]. For these reasons, computer automation seems a necessary component of any long-term monitoring program.

The field of computer vision (CV), particularly deep learning approaches powered by modern GPUs and large annotated datasets (e.g., [27]), has demonstrated the potential for assisting in several laborious visual tasks across different application areas (e.g., agriculture [28], construction [29], and medicine [30]). With the popularization of commercial satellites [31], extremely large amounts of remote sensing imagery are amassed on a daily basis. In order to convert these imagery into actionable insights, CV solutions for remote sensing imagery have become commonplace (e.g., sea ice segmentation and classification [32–34]). In niche applications such as detecting wildlife, however, the lack of expert annotators to create large training datasets imposes a barrier to successfully applying such methods [35]. Nonetheless, there have been several proof-of-concept studies on the applicability of CV on VHR to detect/measure whales [20], pack-ice seals [10], and penguin colonies [36]. Among the difficulties of moving from proof-of-concept studies on wildlife detection in VHR to fully automated survey programs are ensuring that conditions at which detection models are trained and validated capture the full range of scenarios encountered when making predictions.

Here, we present a human-level, fully automated solution to detect pack-ice seals in VHR imagery, built upon our previous proof-of-concept study [10]. Our pipeline works by pre-processing Antarctic coastline VHR imagery through a sea ice segmentation model [32] that narrows down candidate imagery to scenes with relevant sea ice substrate. These selected scenes are scanned for seals using an ensemble of CNN models and ultimately converted to a database of geolocated predicted seals. Models in our pipeline were trained on a large annotated dataset obtained and curated over the course of five years. This

extensive dataset and a massive deployment of GPU resources allowed us to train, validate, and test a wide range of different model solutions. Using annotations from an experienced observer as the basis for comparison, our automated approach outperforms two human observers with >100 h of experience when faced with novel VHR imagery randomly sampled from the existing imagery collection.

## 2. Materials and Methods

Our seal detection pipeline employs an ensemble of CNN models to derive georeferenced seal locations from very-high-resolution satellite imagery (Figure 1). Detailed information on our imagery datasets, CNN model training, and ensembling techniques can be found in the sections below.

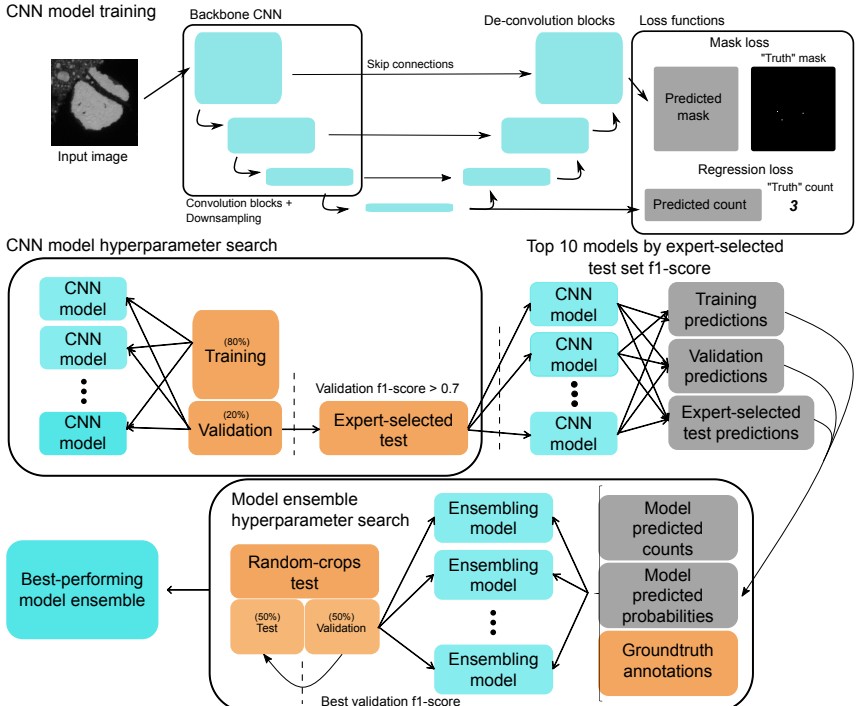

**Figure 1.** Simplified diagram for SealNet 2.0 showing the training of individual CNN models, model hyperparameter search, and model ensembling. Boxes colored in light-blue denote models, orange boxes denote datasets, and gray boxes denote model output. Thick black lines from datasets to models indicate model training. Dashed vertical lines indicate model selection steps. The best individual CNN models are trained on seal detection, including centroid segmentation and seal count regression, using a random search with training and validation, and the f1-score at the expert-selected test set for model selection. The best ensembling models are selected via Bayesian optimization, using top-10 CNN model predictions for the training set, validation set, and the expert-selected test set as dependent variables for training; true positive vs. false positive as the response variable; and the f1-score at the validation split from the random crops test set as a validation metric. Finally, we use the test portion of the random crops test set to estimate the out-of-sample performance of the best-performing model ensemble.

### 2.1. Imagery and Data Annotation

For training, validation, model selection, model ensembling, and out-of-sample performance estimation, we employed three different annotated datasets (Figure 2): a training/validation set, an expert-selected test set, and a randomly selected test set. All three datasets comprised panchromatic Worldview-3 high-resolution satellite images (each one referred to as a 'scene') from the Antarctic coastline with an on-nadir resolution of 0.34 m/pixel. The annotation process consisted of manually browsing through Antarctic

coastline imagery at a scale at which individual seals are detectable; at the centroid of each putative seal found, a geolocated point was added to a GIS (geographic information system) spatial point database (i.e., ESRI ® shapefile). We used a double-observer approach to create a consensus test dataset for out-of-sample performance estimation and model selection. The training/validation and test datasets used in this study were extracted from a set of 38 panchromatic scenes covering 8719.82 km². We selected training and test scenes that would represent a comprehensive range of environmental and sensor conditions, including images captured over a range of off-nadir angles, lighting conditions, and cloud covers. Our training/validation and test sets represent a significant expansion and update of the datasets used in [10]. In addition to eliminating any putative seals that were re-classified on further consideration, we annotated several new scenes for training and validation. We also changed our hard-negative sampling strategy from extracting crops at polygons that marked locations without seals to extracting random crops that did not overlap with any seal annotations in scenes for which we had seal annotations. This new hard-negative sampling approach reflects the introduction of a sea ice detection step prior to detecting seals, allowing us to focus on areas with sea ice conditions amenable to seals being hauled out and thus available for detection. We revised all hard-negative samples to remove potential false positives. To avoid a potential selection bias, we created an additional test set (heretofore, the 'random crops test set') composed of 300 non-overlapping 1 km² crops from 100 randomly selected WV03 scenes from the Antarctic coastline identified as having at least 5% sea ice cover by a sea ice segmentation CNN [32]. These crops were annotated by three different observers (BG, MW, HJL) with varying levels of experience classifying seals in WV03 imagery.

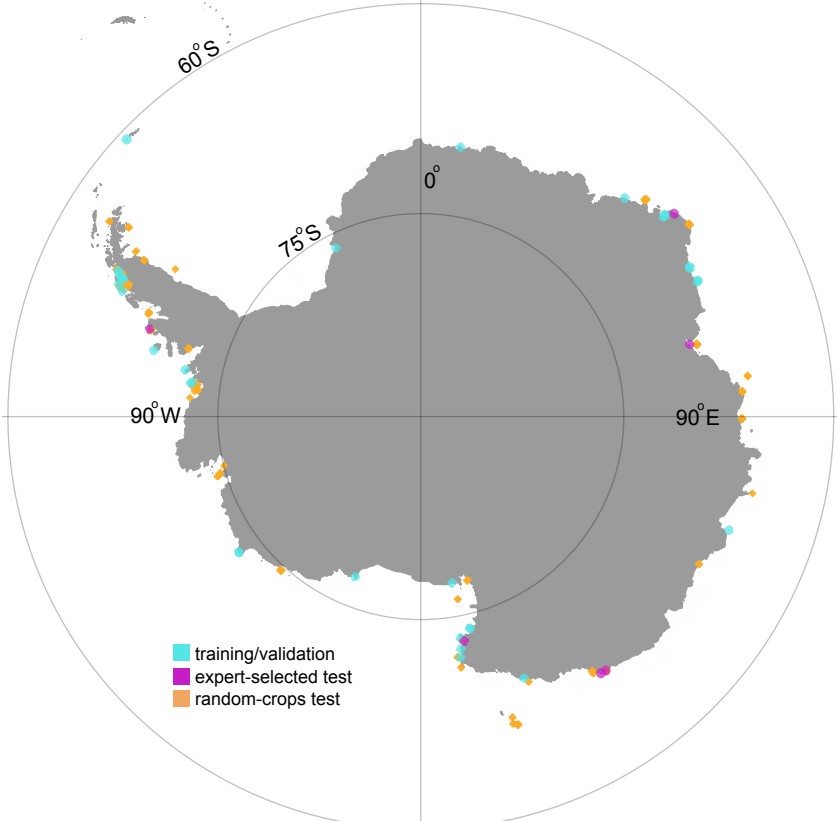

**Figure 2.** Training/validation set (light-blue), expert-selected test set (magenta), and random-crop test set (orange). Polygons denote entire Worldview-3 panchromatic scenes for the training/validation set and the expert-selected test set, and 1 km² crops within panchromatic Worldview-3 scenes for the random-crop test set.

### 2.1.1. Training/Validation Set

We employed non-overlapping training and validation sets to train seal detection CNNs via gradient descent and run a comprehensive hyperparameter search optimized for out-of-sample performance. Using non-overlapping groups of seals from scenes in the training/validation set, we randomly assigned 80% of those groups of seal annotations to model training and 20% to model validation. Centered on each seal location, we extracted a 768 × 768 cropped image (i.e., patch) and saved a binary mask for that crop with '1' on pixels at seal centroids and '0' elsewhere. In addition, we recorded the total number of seals present in each patch. Since several seals may overlap in a single training patch, sampling training images at random would result in a bias towards seals situated within groups of seals (which are easier to detect [10]). To address that, we designed a weighted sampler that ensures that every seal is equally likely to be represented during training by down-weighting the probability of sampling individual seals based on the number of seals found within a radius of 50 m, making solitary seals and seals in larger groups just as likely to be represented during training. For training models that require bounding box annotations (i.e., instance segmentation and object detection models), we generated 11 × 11 bounding boxes centered on each seal. For each scene in training/validation sets, we extracted 300 non-overlapping 'hard-negative' 768 × 768 patches by randomly drawing crops from regions without any seal annotation. To ensure that no false positives existed in negative patches, we manually reviewed each negative patch and excluded those potentially containing seals. The final training/validation set was composed of a total of 8735 patches with seal annotations and 7750 hard-negative patches.

### 2.1.2. Expert-Selected Test Set

Our expert-selection test set builds on the approach from our previous study [10] by adding annotation revisions and replacing the original negative scenes with scenes that better reflect our new pipeline design with its sea-ice detection pre-processing step. We employed scene-wide annotations here to provide realistic out-of-sample metrics of model predictive performance in production (i.e., when predicting, a detection model has to go through entire scenes). This test set was used for both CNN model selection and model ensemble training. Moreover, we used predictions from the 10 top-performing models in the test scenes along with consensus annotations to train model ensembles that reclassify each point as a seal or a false positive based on the prediction from each model for that specific point (see below).

### 2.1.3. Random Crops Test Set

We used a set of 300 non-overlapping 1 km$^2$ crops from 100 randomly selected WV03 scenes to validate models, optimize model ensembles, and remove selection bias from out-of-sample performance estimates. The 100 WV03 scenes were sampled at random from a set of 1948 scenes obtained by classifying a suite of 14,872 WV03 scenes through our sea ice segmentation model [32] and eliminating from consideration those with <5% (predicted) sea ice cover. Because the vast majority of imagery contains neither seals nor features such as rocks and/or shadows that could be confounded with seals, we used our top-10 model ensemble along with a stratified sampling approach to select the 300 crops for the test set; specifically, we selected 100 crops where all 10 models predicted one or more seals (seals very likely present), 100 crops where none of the models predicted any seals (seals very likely absent), and 100 crops where there was disagreement among the models as to the presence of seals in the scene (seal presence ambiguous). To create a consensus dataset representing manual (human) annotation, three observers (BG, MW, and HJL) independently annotated these 300 crops. While all three observers had considerable experience annotating seals in WV03 imagery, there was a gradient in the amount of experience ($BG > MW > HJL$); so, the consensus annotation used to represent 'truth' (in the absence of true ground validation, which is impossible in this scenario) was constructed by having the most experienced observer (BG) review and edit (as appropriate) the union

of all manual annotations and high-probability CNN predictions. To explore the benefits of AI-guided annotation, each observer had a 50% chance of having access to AI help in the form of ensemble model predictions with their associated probability. Finally, we used the random crops test set as a tool to calibrate and evaluate model ensembles on out-of-sample annotations.

### 2.2. CNN Training and Validation

CNN-model architecture choice focused on established rather than state-of-the-art methods to favor explainability, comparison with other studies, and ease of implementation instead of pure predictive performance. Seal detection CNNs tested in this study were designed for three different tasks: object detection (i.e., drawing bounding boxes around each object of interest and giving appropriate labels to each bounding box), instance-segmentation (i.e., object detection with segmentation masks inside bounding boxes), and semantic segmentation (i.e., labeling every pixel in the image). For object detection and instance segmentation, we tested Fast R-CNN [37] and MaskRCNN [38], respectively, as implemented in the native torchvision package [39]. In both cases, we modify the default anchor box sizes for predicted objects to a smaller size that better matches our 'truth' bounding boxes, and, in the case of MaskRCNN, we swapped the original binary cross-entropy (BCE) criterion from the segmentation loss for a region-based dice loss (see the section on loss functions below). For semantic segmentation models, we tested both the U-Net architecture [40], as implemented in the segmentation-models-pytorch package [41] but with an added regression head, and TransUnet, a transformer-based U-Net-like architecture, as implemented in the original paper [42], with the exception of an added regression head. To ensure a fair comparison between widely different CNN modalities, we used a unified validation metric across all models: f1-score between predicted and 'truth' seals in the validation set. Training epochs consisted of going through every training image exactly once, starting a validation round whenever 1/3 of the training images were processed, with a total of three validation rounds per training epoch. CNNs were trained with an AdamW optimizer [43] with a policy that reduces the learning rate by a factor of two whenever there is no improvement in terms of validation metrics for N consecutive validation rounds—where 'N' is a hyperparameter—and stops training whenever there is no improvement for 3 training epochs. Across all settings, we used $512 \times 512$ input images, either sampled at random from the training pool or processed sequentially from the validation pool, grouped into mini-batches with the greatest number of images allowed given constraints from the model architecture and GPU memory availability. All experiments were performed using pytorch [44] with mixed-precision training [45] on NVIDIA V100 GPUs with 32 GB of memory from the Bridges-2 supercomputer [46].

#### 2.2.1. Data Augmentation

To make our models more robust to scale, positioning, illumination, and other potential confounding factors, we employed a data augmentation pipeline during training, tailored to take full advantage of random crops and rotations given the nature of our imagery and dimensions of our objects of interest. We intentionally extracted larger patches ($768 \times 768$) than our model input size ($512 \times 512$) to keep training images diverse and non-obvious, and to reduce a potential bias for detecting seals at the center of input images. Whenever a seal was no longer present in the original patch after applying a random crop, the count for that patch (based on the 'truth' dataset) was adjusted to reflect that. For training, we used two data augmentation strategies: (1) a simple approach with random crops, vertical and horizontal flips, random shifts in position, random re-scaling, random 90-degree rotations (i.e., 90, 180, or 270 degrees), and brightness and contrast shifts; (2) a more complex approach using the transforms listed above plus noise reduction and intensity shifts. Our training augmentations are integrated into the training loop using implementations from the Albumentations package [47]. As an additional step to make predictions more robust

to orientation, we apply horizontal flips and 90, 180, and 270 degree rotations to each input image, and average out predicted masks and counts from all possible combinations.

### 2.2.2. Loss Functions

To train regression heads on semantic segmentation models (i.e., U-Net and TransUnet), we used Huber loss [48], whereas segmentation heads used either Focal loss [49], Dice loss [50], or a combination of both. For instance segmentation and object detection models (i.e., Fast R-CNN and Mask R-CNN), we used the default torchvision losses for the Region Proposal Network classifier, the Bounding box classifier, and the Bounding box coordinates regression. We swapped the original Mask R-CNN BCE loss for predicted masks with Dice loss since BCE is not suitable when few if any pixels fall into the positive class.

### 2.3. Hyperparameter Search and Model Selection

We tested a wide range of scenarios to find optimal combinations of hyperparameters for seal detection models according to f1-score [50] for the expert-selected test set (i.e., test f1-score), running a total of 1056 full-length experiments. For all models, we tested the impact of learning rate, the number of epochs without improvement that would trigger learning rate reduction, and the ratio of negative to positive images in the data loader. For semantic segmentation models, we tested the impact of the segmentation loss function (Dice loss, Focal loss, or mixed Dice and Focal losses), backbone architecture (Resnet34 [51] and EfficentNet [52] variants), relative weight for regression and segmentation losses, and rate of dropout [53] applied to regression heads. When measuring test f1-score for semantic segmentation models, we tested the potential of using a threshold on predicted counts to remove false positives. To analyze the relative impact of each hyperparameter on test f1-score, we fit a CatBoost regressor model using hyperparameter values from each trial as dependent variables and f1-score as the outcome variable and calculated the relative importance of each hyperparameter using Shapley [54] scores. To save processing time, experiments that underperformed in terms of the maximum validation f1-score (>0.7 for instance segmentation and >0.5 for object detection and instance segmentation) were not carried into the testing stage. After an initial set of 372 experiments with semantic segmentation models, we narrowed down our hyperparameter pool to speed up convergence. For the latter 251 experiments, we also used a range of thresholds to explore the impact of using predicted count as a post-processing step to remove false positives, i.e., for each threshold, predicted points on patches where the predicted count was smaller than the threshold were discarded before comparison with ground truth annotations.

### 2.4. Model Ensembling

Our examination of a large suite of models allowed us to deploy a model ensemble post-processing step. The first step to creating model ensembles was gathering the correspondent predicted counts and logits for predicted seal locations at the expert-selected test set and training/validation set using the 10 top-performing models in terms of test f1-score. Whenever an individual model did not predict a seal at a location where other model(s) did predict a seal, cells with model logits and counts for that location were left as missing values. Predicted counts and logits from each model were then used as dependent variables to predict whether each point was in fact a true seal according to the 'truth' annotations. We split our random crops test set between validation and testing to run a hyperparameter search for ensemble models, trained at binary classification for true-positive vs. false-positive seals, ranging from simpler linear models (i.e., logistic regression and ElasticNet [55]) to more intricate tree-based models (i.e., random forest, CatBoost [56], and XGBoost [57]). Though the training and validation sets are already captured by individual models, we added the potential usage of these annotations to train model ensembles as a hyperparameter. We ran 50 independent hyperparameter search studies with 1500 experiments each. An experiment in our hyperparameter search consisted of sampling a combination of hyperparameter values from posteriors, training an ensemble model using those hyperparameter values,

and updating posteriors according to f1-score in the validation portion of the random crops test set. To measure the contribution of each individual model to ensemble predictions, we used relative feature importance for logits and predicted counts from each model, in the form of feature weights for linear models and Shapley scores for tree-based models. We used a Bayesian optimization routine [58], implemented in the optuna package [59], with multivariate normal priors for hyperparameters to find the best-performing ensemble model, using the f1-score of the validation half of the random crops test set as our metric. The full range of hyperparameter choices for ensemble models can be found in the project's code repository.

### 2.5. Evaluation

During all training experiments, our validation metric is the instance-level (i.e., individual seal centroids) f1-score. We measured this metric directly for model ensemble predictions; however, for CNN outputs, we need to match each predicted seal with consensus annotations. Because different annotators may identify the seal centroid in slightly different locations, and given expected seal dimensions of roughly 2 m, we used a tolerance of 1.5 m to declare two seals a match. For semantic segmentation models, we did so by applying a sigmoid transform followed by a binary thresholding step, leaving us with seal mask polygons. We then extracted the centroid of each polygon and looked for a match with the centroids of the consensus dataset. For instance segmentation and object detection models, however, we simply extracted the centroid from each predicted bounding box for comparison with consensus centroids. We evaluated the out-of-sample performance of model ensemble predictions, the best individual performing models, the original SealNet model [10], and human observers against our random crops test set consensus annotations. Since model ensembles use the validation portion of the random crops test set for model selection, all models and model ensembles were evaluated on the test portion of the random crops dataset to ensure a fair comparison. For all evaluation steps, model predictions on land were masked out using a sea ice mask derived from the Antarctic Digital Database (ADD) high-resolution coastline polygons available on the Quantarctica project [60]. To evaluate the consistency of output model probabilities, we measured the correlation between the sum of logits around predicted seal centroids and their corresponding 'truth' label. Similarly, we measured the same correlation for ensemble models using model-derived logits and their corresponding 'truth' labels.

## 3. Results

Semantic segmentation models largely outperformed object detection and instance segmentation models (Figure 3) in terms of test f1-score ($0.39 \pm 0.08$, $0.04 \pm 0.02$, and $0.04 \pm 0.02$, respectively), attaining a top test f1-score of 0.58. Our initial set of experiments with semantic segmentation models (Figure 4, marked in orange) showed that test-time-augmentation is beneficial in terms of test f1-score, and that mid-range backbone architectures in terms of complexity (i.e., EfficientNet-b0, b1, and b2) had a slight edge over the extremes (i.e., ResNet34 and EfficientNet-b3); thus, we turned test-time-augmentation on by default and focused on mid-range backbone architectures for the later part of hyperparameter search experiments. First phase results also hinted that our ranges for the ratio of negative to positive images in training batches, learning rate, regression head weight, regression head dropout, and learning rate scheduler patience (Figure 4, marked in orange) could be adjusted to speed up convergence on better-performing models. Though there was a slight edge for simple augmentations over complex augmentations and random sampling over weighted group sampling, we opted to keep both options for later experiments. Even with fewer iterations, final experiments (Figure 4, marked in teal) largely outperformed initial ones in terms of test f1-score ($0.35 \pm 0.09$ vs. $0.26 \pm 0.13$). Using a threshold on predicted counts as post-processing to remove false positives showed an average increase of $0.04 \pm 0.05$ in terms of test f1-score. Applying a threshold based on predicted counts as a post-processing step dramatically changed the distribution of test

f1-scores (Figure 5); moreover, virtually all best-performing models had increases in f1-score by applying post-processing via the predicted count threshold.

The best-performing study for ensemble models achieved an f1-score of 0.69 in the validation portion of the random crops tests set (Table 1). The correlation between model logits and 'truth' labels was consistently higher for ensemble models when compared with individual CNN models (Table 1). The vast majority of hyperparameter search studies (42 out of 50, binomial *p*-value: <0.001) converged on XGBoost tree ensembles as their model of choice, with varying internal settings. The single best-performing study and 2 out of the 10 best-performing studies, however, converged on CatBoost tree ensemble as their model of choice. All independent studies kept CNN model predictions from the validation set as training data and dropped those for the training set. Shapley values for feature importance on the best-performing models in independent studies (Figure 6) heavily favored features from the best-performing CNN in terms of f1-score in the random crops test set (CNN 3) and features regarding patch-level predicted counts, followed closely by model logits.

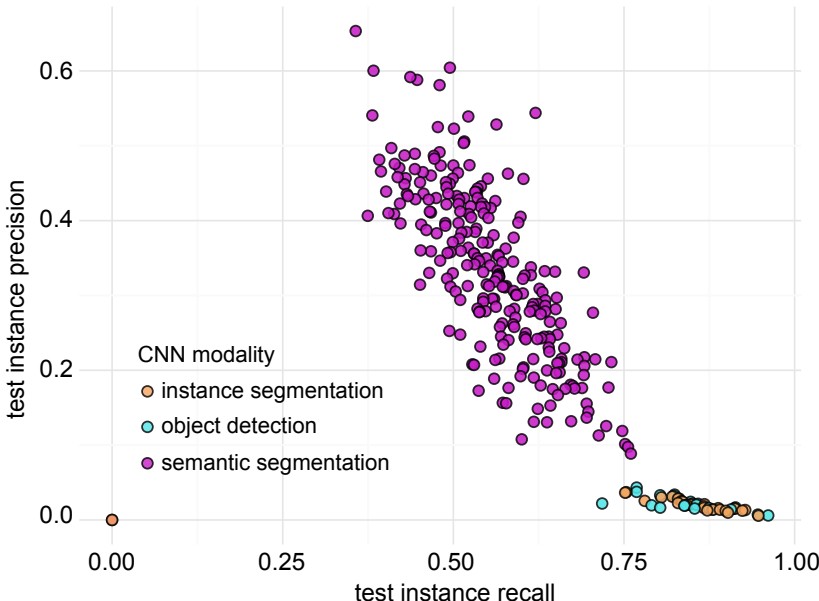

**Figure 3.** Expert-selected test set f1-score for hyperparameter search experiments from different computer vision domains. To ensure a fair comparison of models from different domains, semantic segmentation output masks are passed through a sigmoid transform and thresholded to extract mask centroids. Similarly, instance segmentation and object detection output bounding boxes are converted to centroids to evaluate matches with 'truth' centroids. To avoid unnecessary expenditure of GPU credits, experiments that did not perform well on the validation set (validation f1-score > 0.7 for semantic segmentation models and >0.5 for instance segmentation and object detection models) were not carried into testing.

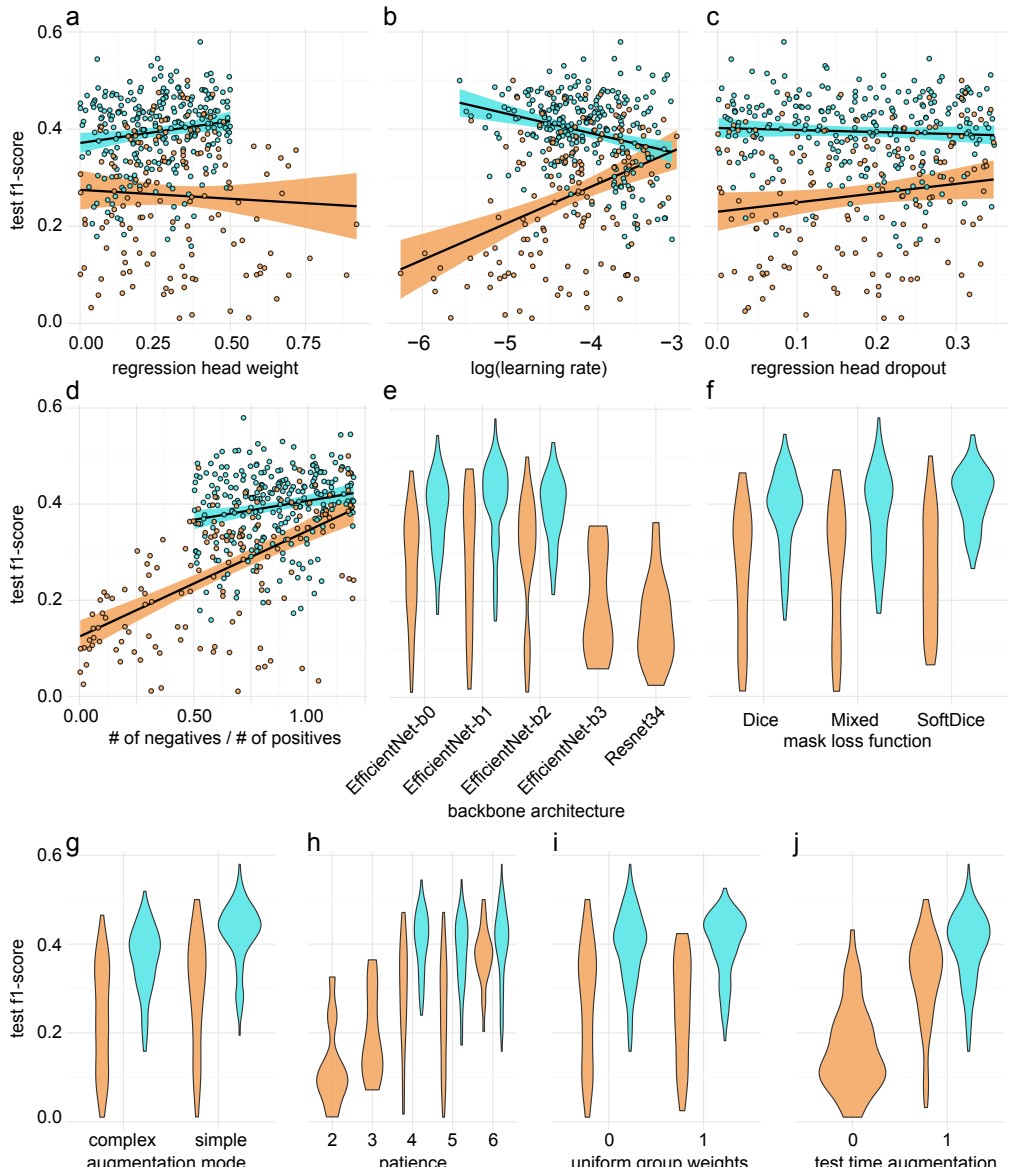

**Figure 4.** Results from random search hyperparameter study for semantic segmentation models, with phases one (orange) and two (teal). For continuous hyperparameters—namely, regression head weight, learning rate, regression head dropout, and negative-to-positive ratio—each circle corresponds to an independent random search experiment. For the continuous hyperparameters—regression head weight, learning rate, negative-to-positive ratio, and the discrete parameters—backbone architecture, patience, and test time augmentation, we narrowed down the range of options to speed up convergence on a best-performing model. Apart from narrowing down hyperparameter priors, we also used phase 2 to test the potential benefit of using regression output as a post-processing step to remove potential false positives, and thus, the test f1-score for those experiments is calculated after post-processing. The width of violin shapes for categorical variables (panels **e**–**j**) indicate the proportion of observations that take on those values, whereas X-axis values indicate all discrete states tested for that categorical variable. Phase 2 experiments tested a narrower range of discrete states for test time augmentation, patience, and backbone architecture; thus, there were no phase 2 results for some discrete states of those variables. Experiments for which test f1-score was below 0.01 are excluded from this plot.

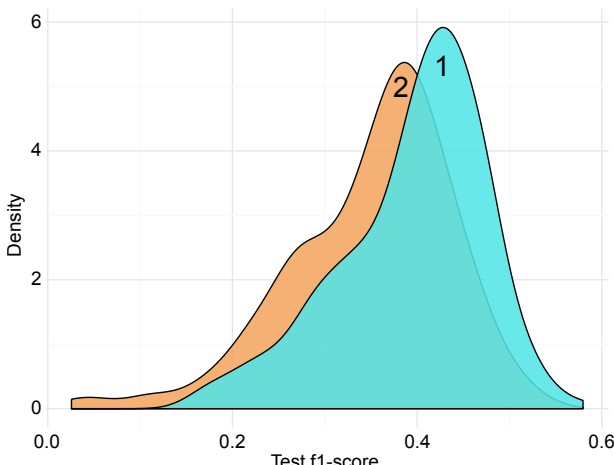

**Figure 5.** Side-by-side comparison between phase 2 experiment results with (1) and without (2) regression post-processing. Post-processing consisted of discarding predicted points within patches where regression output (i.e., predicted number of seals in patch) is smaller than a specified threshold. For each model, we explore a range of thresholds to obtain the maximum possible test f1-score obtained after post-processing, using the same optimal threshold across the entire expert-selected test set.

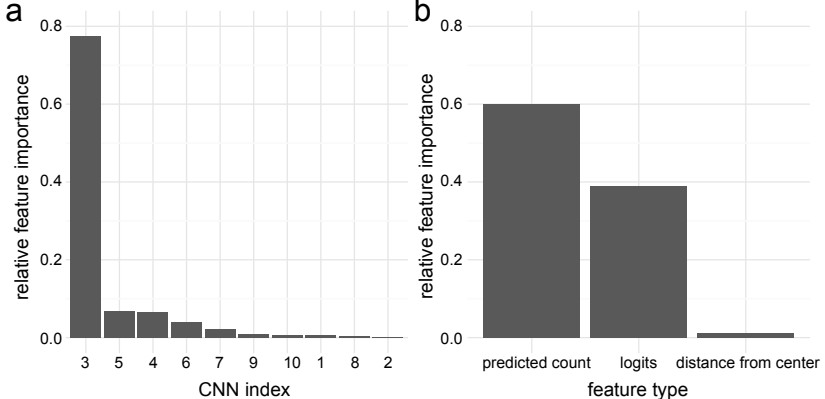

**Figure 6.** Feature importance for ensemble model features grouped by CNN index (**a**) and feature type (**b**). Ensemble models were either CatBoost or XGBoost tree-based ensembles trained for classifying false-positive and true-positive seal detections. Models were derived from a hyperparameter search using a training set with the logits, predicted seal counts, and distances from crop centers from the output of 10 U-Net CNNs at the validation and expert-selected test sets. Feature importances were obtained via Shapley scores at the validation portion of the random crops test set.

**Table 1.** Out of sample performance for human observers (with and without the help of AI output), individual CNN models, and model ensembles measured at the random crops tests set. AI help is provided through the output of a simple ensemble model (i.e., an ElasticNet classifier, 'ensemble naive'), with a color gradient based on model certainty. Because whether an observer will have access to AI help is assigned independently at random, human observers had different sets of imagery processed with the aid of AI output. U-Nets 1–5 are ordered according to their ranking based on f1-score in the expert-selected test set. SealNet 1.0 predictions were obtained with the original SealNet. Similarly, ensemble models 1–5 are numbered in descending order of f1-score on the validation portion of the random crops test set. We include the correlation between model logits and 'truth' labels as a measurement of consistency.

| Observer/Model | Precision | Recall | f1 | AI Help | Architecture | Logit Correlation |
| --- | --- | --- | --- | --- | --- | --- |
| HJL | 0.35 | 0.56 | 0.43 | No | - | - |
| HJL | 0.58 | 0.69 | 0.63 | Yes | - | - |
| MW | 0.50 | 0.63 | 0.56 | No | - | - |
| MW | 0.55 | 0.69 | 0.61 | Yes | - | - |
| CNN 1 | 0.60 | 0.63 | 0.62 | - | UnetEfficientNet-b1 | 0.54 |
| CNN 2 | 0.45 | 0.67 | 0.54 | - | UnetEfficientNet-b1 | 0.33 |
| CNN 3 | 0.71 | 0.67 | 0.69 | - | UnetEfficientNet-b1 | 0.60 |
| CNN 4 | 0.44 | 0.67 | 0.53 | - | UnetEfficientNet-b1 | 0.36 |
| CNN 5 | 0.68 | 0.53 | 0.60 | - | UnetEfficientNet-b0 | 0.53 |
| SealNet 1.0 | 0.07 | 0.02 | 0.03 | - | SealNet | 0.07 |
| ensemble 1 | 0.80 | 0.64 | 0.71 | - | CatBoost | 0.69 |
| ensemble 2 | 0.74 | 0.67 | 0.70 | - | XGBoost | 0.67 |
| ensemble 3 | 0.64 | 0.70 | 0.67 | - | CatBoost | 0.67 |
| ensemble 4 | 0.73 | 0.67 | 0.70 | - | XGBoost | 0.68 |
| ensemble 5 | 0.73 | 0.66 | 0.70 | - | XGBoost | 0.67 |
| ensemble naive | 0.59 | 0.69 | 0.64 | - | ElasticNet | 0.60 |

## 4. Discussion

Our best-fitting CNN ensembles (Figure 1) attain an f1-score of 0.71 on a randomly-sampled dataset, with double-observer coverage and no exposure during training or validation, outperforming two human observers with >100 h of experience and access to annotations from both a simpler ensemble (Table 1) and the previous SealNet CNN pipeline [10]. The improvement in predictive performance stems from three primary factors: (1) a larger and more carefully curated training dataset that focused on scenes with sea ice (Figure 2); (2) a comprehensive hyperparameter search study (Figures 2 and 4), only feasible in a multi-GPU setting; and (3) a new methodology using binary thresholding and regression counts followed by a model ensemble post-processing step. Our SealNet 1.0 classifier used the regression output to dictate how many logit hotspots would be extracted from predicted segmentation masks [10]. Here, we use regression outputs as a post-processing step for segmentation masks, which removes many false positives and leads to an improved f1-score (Figure 5). This SealNet 2.0 approach is also preferable because it relies solely on pixel-level, centroid mask annotations for prediction, which hinge upon stronger supervision signals during training when compared with patch-level 'true' counts.

Surprisingly, though the problem at hand theoretically aligns better with instance segmentation/object detection frameworks, our experiments with MaskRCNN [38] and Fast R-CNN [37] showed lackluster results (Figure 3) when compared with U-Nets [40]—a considerably simpler semantic segmentation approach. The extremely poor precision scores obtained with these methods could derive from limitations for training without foreground objects, creating a bias for over-predicting seals. With our U-Net-based approach, we are not only able to train using background-only patches, but we can also find optimal ratios of patches with and without foreground objects to maximize the balance between precision and recall through a hyperparameter search (Figure 4). This capability could give an edge to U-Net-based and other semantic segmentation approaches in 'needle-in-a-haystack'

problems, which are ubiquitous in object detection applications for remote sensing imagery (e.g., [20,61,62]). The importance of showing negative examples during training in this kind of setting is supported by the relatively high negative-to-positive ratio found in our best-performing models (Figure 4, panel d).

Our top-10 individual CNN models, surprisingly, have a slightly lower out-of-sample recall than the global average for phase 2 experiments (0.54 vs. 0.55); however, on average, they are able to attain dramatically higher precision (0.52 vs. 0.33). This emphasis on avoiding false positives is also present when we look at the extremely high correlation between out-of-sample precision and f1-score ($r = 0.93$) and the strong negative correlation between out-of-sample recall and f1-score ($r = -0.43$). The relatively low correlation between f1-scores on the expert-selected test set performance and the random crops test set ($r = 0.49$)—and the dramatic performance decrease from SealNet 1.0 [10] on our more diverse test set Table 1—illustrates the importance of designing comprehensive test suites and cautions against over-relying on performance estimates on limited test sets.

Having access to AI help in the form of output from a simple ensemble model leads to a substantial improvement in the f1-score of human observers, improving precision without sacrificing recall. Though we are not able to draw statistical insights given our limited observer pool, our results suggest that human supervision could be used as quality control for AI output, as in most human-in-the-loop AI approaches (HITL [63,64]), and may also be used to guide inexperienced observers on challenging detection/classification tasks such as ours.

Though our ensemble models consistently outperformed individual CNN models (Table 1), we found the performance boost to be too small to justify the added computational cost of running imagery through ensembles when compared with the best-performing individual CNN (CNN 3, f1-score 0.69 vs. ensemble 1, f1-score 0.71). This similarity in performance is not surprising given the pronounced impact of features coming from CNN 3 on the best-performing ensembles (Figure 6), with minor contributions from a few other CNN models. Moreover, though we had a diverse set of hyperparameters within our 10 best-performing models, they hinged on the same datasets and model architecture (U-Net), which may have contributed to the high redundancy in including features from multiple CNNs. In contrast, several successful cases of applying ensemble models to CV rely on merging widely different individual components (e.g., [65,66]). On the other hand, ensemble models consistently outperformed individual CNNs in terms of the correlation between model logits and true labels (Table 1), which makes them more desirable as an AI-guided annotation tool and could translate to a lower bias on novel input imagery.

While our approach performs extremely well in simple terrain, with large groups of seals, it does encounter difficulties with rough terrain and single seals (Figure 7). Notably, these settings also tend to be the most challenging for human observers, shown by the high correlation between AI and observer errors. These difficulties are unlikely to be surmounted directly by improvements in AI because we cannot reliable annotate ground-truth datasets in these settings. Although identifying the portions of VHR scenes where seals, if present, could be detected is a tractable problem for modern CV models, estimating seal densities in areas where they cannot be detected is a non-trivial problem and merits further investigation.

In addition to high out-of-sample performance (Table 1), when compared with other candidate sampling methods for surveying pack-ice seals (i.e., fixed-wing airplanes, helicopters, UAVs, and human expert-based VHR surveys), AI-based approaches showed an operational cost comparable to that of the cheapest option available (helicopter-based flight transect surveys) [26], even taking into account the considerable cost of purchasing commercial VHR imagery. Cost-efficiency aside, GPU-accelerated AI-based approaches generate orders of magnitude less emissions than any other surveying method mentioned above [26]. The success of citizen science campaigns such as SOS [24], however, show the potential to utilize regular citizen science surveys as a validation method for fully

automated pack-ice seal detection pipelines, especially given the difficulty of covering every potential real-world scenario during model evaluation.

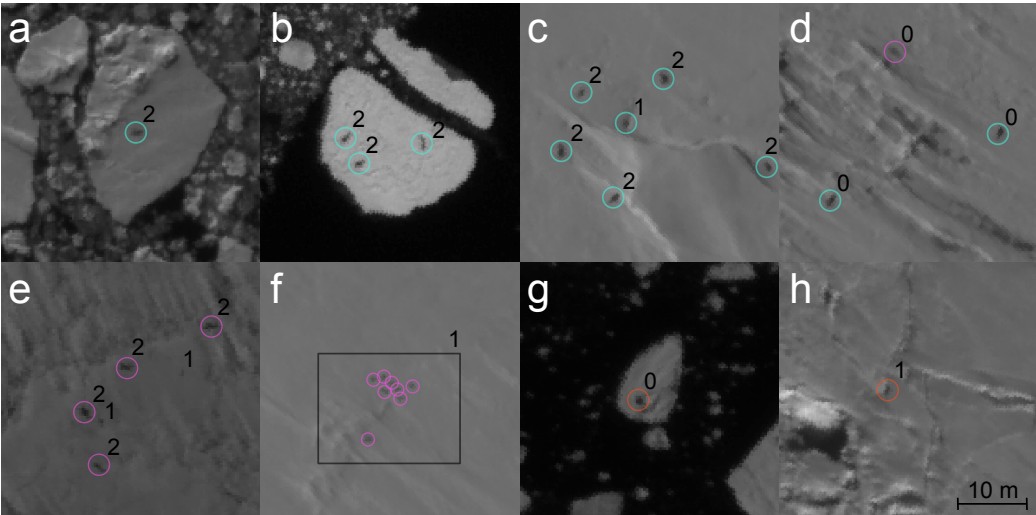

**Figure 7.** Prediction samples from our best ensemble model on eight scenes from the random crops test set. Samples were chosen to represent scenarios where the model predicts seal locations correctly (panels **a–d**), fails to find existing seals (panels **d–f**), and annotates background objects as seals (panels **g,h**). Seals marked with teal circles indicate true positives (i.e., predicted seal present in consensus dataset), whereas purple and orange circles indicate false-negative and false-positive seals, respectively. Numbers next to circle annotations indicate the number of human observers that agreed with that particular model annotation. Number '1' annotations unaccompanied by circles in panel **e** indicate edge cases where a human observer annotated a seal that was not present in the consensus dataset or model predictions. Imagery copyright Maxar Technologies Inc. 2022.

## 5. Conclusions

Our results show compelling evidence for the immediate applicability of CNN-based, fully automated approaches for pack-ice seal surveys in VHR imagery. Moreover, they highlight the importance of comprehensive hyperparameter search studies and diverse training and evaluation datasets when employing AI methods to address complex tasks such as Antarctic pack-seal annotation in VHR imagery. With the addition of a pre-processing step to select VHR scenes where seals, if present, could be found and regular random checks by human observers for quality control, our approach with CNN ensembles is capable of delivering reliable, continental-scale putative Antarctic pack-ice seal locations.

**Author Contributions:** Conceptualization, B.C.G. and H.J.L.; methodology, B.C.G.; software, B.C.G.; validation, B.C.G., H.J.L. and M.W.; data curation, B.C.G., H.J.L. and M.W.; writing—original draft preparation, B.C.G.; writing—review and editing, B.C.G., M.W. and H.J.L. All authors have read and agreed to the published version of the manuscript.

**Funding:** This research was funded by the U.S. National Science Foundation (Award 1740595) and the U.S. National Aeronautics and Space Administration (Award 80NSSC21K1150). Geospatial support for this work provided by the Polar Geospatial Center under NSF-OPP awards 1043681 and 1559691.

**Data Availability Statement:** Annotation datasets with geolocated seals for training, validation, and test sets, along with a document code repository for model training, validation, and prediction, are available at the project's GitHub page, accessed on 20 September 2022.

**Acknowledgments:** We acknowledge support and computational resources provided by the Stony Brook University Institute for Advanced Computational Sciences, the Rutgers RADICAL group, and staff of the Bridges-2 computing program.

**Conflicts of Interest:** The authors declare no conflict of interest.

**Abbreviations**

The following abbreviations are used in this manuscript:

| | |
|---|---|
| VHR | very-high-resolution (satellite imagery) |
| SO | Southern Ocean |
| CV | computer vision |
| GIS | geographic information system |
| ADD | Antarctic Digital Database |
| CNN | convolutional neural network |

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
