# Peer review of "SealNet 2.0: Human-Level Fully-Automated Pack-Ice Seal Detection in Very-High-Resolution Satellite Imagery with CNN Model Ensembles"

_remotesensing, doi:10.3390/rs14225655_

Round 1

Reviewer 1 Report

This manuscript focuses on pack-ice seal detection in very-high-resolution satellite imagery with CNN model ensembles. There are many defects in it.
1.In line 4, "SeaNet2" should be replaced by "SealNet 2.0".
2.In line 10, what is "HPC"?
3.In lines 100-101, the sentence "each one a collection of geographic point locations" is strange.
4.In line 107, what is "GIS"?
5.In line 107, what is "ESRI"?
6.In lines 188 and 360, "FastRCNN" should be replaced by "Fast R-CNN".
7.In lines 189, 191, and 359, "MaskRCNN" should be replaced by "Mask R-CNN".
8.In line 232, "Faster-R-CNN" should be replaced by "Fast R-CNN".
9.In line 303, what is "ADD"?
10.In lines 310 and 311, "inverse ±" should be replaced by "±".
11.In line 7, "best ensemble attains 0.80 6 precision and 0.64 recall" is mentioned. In Fig. 3, the best results are 0.806 recall and 0.64 precision. The data are different.
12.In line 351, "SealNet1.0" should be replaced by "SealNet 1.0".
13.In Fig. 4(d), the title for the x axis is "negative to positive ratio". But, no negative ratios are shown.
14.In Figs. 4(e), 4(f), 4(g), 4(h), 4(i), and 4(j), what do the widths mean? Or, what are their x axes?
15.In Figs. 4(e), 4(h), and 4(j), why are orange items more than teal items?
16.In Fig. 6(b), what are the y axis and its unit?
17.In Table 1, "F1" should be replaced by "f1".
18.In Table 1, "SealNet 1" should be replaced by "SealNet 1.0".
19.Why are the architecture in Fig. 4(e) different from the architecture in Table 1?
20.In Fig. 7(d), "0" next to the purple circle should be replaced by "1".
21.In Fig. 7(e), there are four circles and five numbers.
22.In Fig. 7(h), "1" should be replaced by "0".
In conclusion, major revision is necessary.

Reviewer 2 Report

This manuscript used the ensemble method, which consisted of multiple CNN-based models, to study the pack-ice seal detection issue. This study is interesting and worth to be published, but there are still several issues listed below this reviewer would like the authors to address:

1. In line 138, how did the “weighted sampler” works?

2. Please briefly describe the structures of the top 10 performed models, which experts selected to form ensemble 1, and CNN 3, which gave the best individual model performance.

3. Do you mean “semantic segmentation” instead of “instance segmentation” in line 193 and line 229?

4. In the title of Table 1, what are the U-Nets 1-5? And in table 1, why the performance of your previous version SealNet 1 dramatically dropped?

Reviewer 3 Report

The article is interesting and deals with a current issue. The introduction is correct and the bibliographic references are correct.

The proposed method is interesting and there are some problems that should be reviewed and some experiments that should be studied in more depth.

Even the authors of the article wrote in the conclusions that the proposed method is limited in rugged areas and where there are few seals or they are not in groups.

The authors' research is to be appreciated and in principle I agree that the article should be published, but after some additions and revisions regarding the proposed method and the deepening of some aspects, which would lead to concrete results and the method should be supported from a scientific point of view .

I suggest that the authors take into account other similar works and other specific methods and make an improvement of the proposed method.

Reviewer 4 Report

1. Why choose FastRCNN, MaskRCNN and UNet as the models for object detection and instance segmentation? What was the basis for the selection?

2Is it possible to provide a detailed figure of the CNN network structure? 

3.How to define the network output as truth positive? Is it possible to provide the formula for the evaluation metrics in the Evaluation?

4. What are the characteristics of the eight scenarios selected in Figure 7? Are they representative? 

5. Some future directions should be pointed out in the Discussion.

Round 2

Reviewer 1 Report

This revised manuscript is ready to be published.

Reviewer 3 Report

The article is interesting and deals with a current issue. The introduction is correct and the bibliographic references are correct.

The proposed method is also interesting in its current form modified by the authors according to what was suggested to me.

The authors' research is to be appreciated and in principle I agree that the article should be published, after some additions and revisions have been made regarding the proposed method and the deepening of some aspects, which can lead to concrete results.

The authors cited other similar works and other specific methods and made an improvement of the proposed method.

The article in its current form is suitable to be accepted in the journal and to be published.

I propose that the article be published and congratulate the authors for promptness and solicitude.

Reviewer 4 Report

I think the revised manuscript can be  accepted.